# Impact of Skin Decontamination Wipe Solutions on the Percutaneous Absorption of Polycyclic Aromatic Hydrocarbons

**DOI:** 10.3390/toxics12100716

**Published:** 2024-09-30

**Authors:** Chandler Probert, R. Bryan Ormond, Ronald E. Baynes

**Affiliations:** 1Wilson College of Textiles, North Carolina State University, Raleigh, NC 27606, USA; 2College of Veterinary Medicine, North Carolina State University, Raleigh, NC 27606, USA; rebaynes@ncsu.edu

**Keywords:** skin decontamination, dermal absorption, polycyclic aromatic hydrocarbon, fireground contaminants, skin wipe, firefighter

## Abstract

Firefighter occupational exposures were categorized as a class 1 (known) carcinogen by the International Agency for Research on Cancer in 2022. As a result, firefighters have become heavily focused on identifying effective and easy to implement decontamination strategies to reduce their chemical exposures. Skin decontamination using wipes post-exposure is one decontamination strategy that every firefighter has available to them. However, firefighters have expressed concerns over the ingredients in the wipe solution increasing dermal absorption. The goal of this study was to determine if the ingredients in skin decontamination wipe solution had any enhancement effect on the dermal absorption of phenanthrene. To determine any enhancement effects, the additive solution of four skin decontamination wipe products was applied to porcine skin 15 min after chemical dosing. The absorption of phenanthrene was tested in vitro using a flow-through diffusion cell system over eight hours. The wipe solution effects on dermal absorption were determined by measuring multiple absorption characteristics including cumulative absorption (µg/cm^2^), absorption efficiency (% dose absorbed), lag time (minutes), flux (µg/cm^2^/h), diffusivity (cm^2^/h), and permeability (cm/h). No penetration enhancement effects were observed in any of the skin decontamination wipe solutions tested; rather, all wipe solutions decreased the absorption of phenanthrene. Slight differences in cumulative absorption among two pairings of skin decontamination wipe solutions, wipes 1 and 3 vs. wipes 2 and 4, were observed, indicating that some ingredients may impact dermal absorption. These findings show that firefighters should continue using skin decontamination wipes to reduce their dermal exposures to fireground contaminants with little concern of increasing the absorption of phenanthrene.

## 1. Introduction

Firefighting is an inherently dangerous profession, with firefighters routinely risking their lives to serve the general public. However, the hazards firefighters face go beyond the obvious threats of heat, smoke, and collapsing structures. The fireground is a complex and dynamic environment, where a multitude of materials are burning and releasing a mixture of toxic chemicals. The burning of plastics, foams, electronics, and other petroleum-based and synthetic materials used in modern home furniture, appliances, and insulation produce a significant amount toxic chemicals, smoke, and soot when burned [1,2]. Firefighters are exposed to these toxic and carcinogenic combustion byproducts, which can have severe acute and chronic health consequences [3,4,5,6,7,8,9,10,11].

Among the most concerning contaminants present on the fireground are polycyclic aromatic hydrocarbons (PAHs), a group of chemicals formed during the incomplete combustion of organic materials [1]. Studies have characterized firefighter exposures to polycyclic aromatic hydrocarbons (PAHs) during training exercises, fire response, overhaul, and automobile fires [3,9,12,13,14]. PAH concentrations in air samples have been found to exceed either published ceiling values, short-term exposure limits, or NIOSH recommended exposure limits [7]. Furthermore, PAHs have been found on the gear and skin of firefighters after fire response [3,15].

Acute exposures to PAHs are important for firefighters as they can cause headaches, nausea, vomiting, and respiratory and dermal irritation [1]. However, the chronic exposure to PAHs may be more relevant to firefighter health and cancer risks. Chronic and occupational exposures to PAHs are correlated with increased incidence of lung, skin, and bladder cancers [16,17,18]. Furthermore, the exposure pathway is a key component when considering which type of cancer forms, in both human and animals [1]. Firefighters can protect against the inhalation exposure pathway by wearing their self-contained breathing apparatus; however, firefighters have minimal to no means of protecting against the dermal absorption exposure pathway.

Dermal absorption may occur when a chemical encounters the skin, either through direct contact or transported to the skin by a vehicle (i.e., benzo[a]pyrene on a soot particulate coming in contact with the skin) [19]. To penetrate the skin, contaminants must overcome the resistive barrier of the stratum corneum, the outermost layer of the epidermis [20]. The stratum corneum (SC) is comprised of 10–30 layers of dead skin cells called corneocytes, which are embedded in a lipid matrix [21]. This skin cell and lipid matrix resembles a brick-and-mortar arrangement, which forms a highly organized and dense structure. This provides the SC its barrier properties against most polar and ionic compounds, as well as external agents like bacteria and fungi [20]. However, this barrier is not perfect, and non-polar and lipophilic compounds, such as PAHs, penetrate by diffusing through the lipid domains [22].

PAHs have been repeatedly shown to penetrate the skin [23,24,25,26,27]. When PAHs are absorbed into the body, they bind to aryl hydrocarbon receptors (AHRs) found in common cells. The body then attempts to metabolize these bonded products by increasing the hydrophilicity to excrete the chemical from the body [18,28]. Benzo[a]pyrene (BAP), as an example, is metabolized into benzo[a]pyrene-7,8-diol and then benzo[a]pyrene-7,8-diol-9,10-epoxide, also known as “the ultimate carcinogen”, and reacts with DNA, forming DNA adducts, which cause cancer [17,18].

To mitigate chemical exposures, firefighters are implementing on-scene decontamination strategies to clean the turnout gear as well as themselves. The goal of on-scene decontamination is to remove the bulk of contaminants that may be on the surface of the gear or skin of firefighters after fire scene exposure, thus reducing their chemical exposures [29]. Firefighters often report positive attitudes towards clean turnout gear and believe cleaning their gear will reduce their risk of cancer; however, there is a large disconnect between their beliefs and actions. A 2017 survey (n = 482) reported that nearly 95% of firefighters claimed they never or only occasionally bagged their gear before returning to the fire station. The same study reported that less than 20% of firefighters frequently or always cleaned their gear before leaving the fire scene or used decontamination wipes [29]. Significant challenges such as knowledge on how to perform decontamination cleaning procedures, time, personnel, and cost may prevent smaller or rural fire departments from routinely participating in on-scene decontamination [29,30,31]. On the other hand, the practice of using a skin decontamination wipe has appeared to increase. A 2023 study reported that skin decontamination wipes were the most common decontamination procedure, with over 60% of respondents using wipes to clean exposed areas post-exposure [31]. Unfortunately, the practice of bagging turnout gear and storing it in the cab was the least common decontamination procedure reported, consistent with findings from 2017 [31]. Although there are several decontamination procedures that are recommended by best practices, few are routinely followed by a majority of firefighters [29,30,31]. Skin decontamination wipes are easy to use, store, and transport, and address several challenges that deter firefighters’ ability to partake in some form of on-scene decontamination.

Even though skin decontamination is one of the easiest decontamination strategies, there are limited data on the safety and effectiveness of decontamination wipe products on removing contaminants from the skin. Previous studies of wipe manufacturers have suggested that their decontamination wipes are effective at removing contaminants from the skin. However, the testing surfaces used to simulate the skin are non-porous and are unable to absorb chemicals like human skin [32,33,34]. Additionally, some firefighters have expressed concerns over some ingredients in the skin decontamination wipes that may potentially increase dermal absorption. These concerns are valid; for example, glycerin has been shown to be a penetration enhancer in rats [35]. Furthermore, other skin decontamination wipe ingredients like plant extracts, polysorbate 20, and citric acid may have the potential to serve as a penetration enhancer as well, if concentrations are high enough [36,37,38]. Therefore, investigation into the effect of skin decontamination wipes on dermal absorption is needed.

The impact of skin decontamination practices and dermal absorption of PAHs under fire fighting conditions (elevated skin and core body temperature, sweaty skin, humid environments, etc.) has yet to be studied. The aim of this study was to investigate if the ingredients in the skin decontamination wipe solution had any penetration enhancement effects on the dermal absorption of phenanthrene, a fireground contaminant. Phenanthrene was applied to the surface of porcine skin in vitro using an artificial sweat dosing vehicle. The solution of four different skin decontamination wipe products were extracted and applied to the test skin 15 min after chemical dosing. The results of this study were compared to data from Probert et al. (2024) [39]; manuscript submitted for publication), which published data on the dermal absorption of phenanthrene under identical conditions. These data served to show the absorption of phenanthrene if no skin decontamination was used by a firefighter post-fire exposure. The goal of this study was to improve the understanding of firefighter chemical exposures through dermal absorption and to evaluate the safety of skin decontamination wipes to establish better firefighting protocols and decontamination practices. This was the first study to investigate the dermal penetration enhancement effects of different skin decontamination wipe solutions in vitro with an artificial sweat dosing vehicle to mimic the sweaty skin of firefighters. The data generated from this study will help inform firefighters regarding the safety of using skin decontamination wipes, as well as help wipe manufacturers understand the effects of the ingredients used in their products.

## 2. Materials and Methods

### 2.1. Skin DecontaminationWipe Materials

Four different skin decontamination wipe products were selected to test the effects of the additive solution on dermal absorption. Each wipe product was secured through their respective commercial vendors. Wipe solution was collected from each wipe by wringing it out by hand into a glass vial. The decontamination wipe solution was then administered to the diffusion cells 15 min after chemical dosing. Several ingredients were found in multiple decontamination wipe products, including water, glycerol, plant extracts, caprylyl, tea tree oil, polysorbate 20, phenoxyethanol, sodium benzoate, citric acid, and more. The ingredient lists of the skin decontamination wipes tested in this study can be found in Table 1. Experimental set up can be found in Appendix A.

### 2.2. Chemicals

The test chemical ^14^C-phenanthrene (specific activity = 55 mCi/mmoL) was obtained from American Radiolabeled Chemicals (St. Louis, MO, USA). Artificial eccrine perspiration (pH = 4.5 stabilized with bactericide and fungicide), an artificial sweat, was used as the dosing vehicle and obtained from Pickering Laboratories (Mountain View, CA, USA). The cells used for the flow-through experiment were 9 mm (0.64 cm^2^) in-line diffusion cells obtained from PermeGear (Hellertown, PA, USA). The collection media were made up the day before the experiment and frozen overnight. Ingredients for the collection media included bovine serum albumin fraction V (2.25% *w*/*v*), sodium chloride (0.3% *w*/*v*), potassium chloride (0.018% *w*/*v*), calcium chloride (0.014% *w*/*v*), potassium phosphate monobasic (0.008% *w*/*v*), magnesium sulfate (0.015% *w*/*v*), sodium bicarbonate (0.14% *w*/*v*), dextrose (0.06% *w*/*v*), distilled water (96.94% *v*/*v*), sodium heparin (0.25% *v*/*v*), amikacin (0.0063% *v*/*v*), and penicillin G sodium (0.0025% *v*/*v*) (Millipore Sigma, Burlington, MA, USA).

### 2.3. Flow-Through Diffusion Cell Set Up

The flow-through diffusion cell system, described by Bronaugh and Stewart [40], was used to perfuse porcine skin membranes. Fresh porcine skin was obtained from the dorsal area of Yorkshire/Landrace pigs 20–60 kg in size. Pig skin has been shown to be similar to human skin with respect to stratum corneum, epidermal thickness, and permeability, with less variability [41,42]. The pigs were shaved and dermatomed to a thickness of 200–300 µm with an electric dermatome (Padgett Instruments, Kansas City, MO, USA). Afterwards, each piece of skin was cut into a circular disk, using a biopsy punch, placed into the diffusion cell, and secured in place, providing a dosing surface area of 0.64 cm^2^. The porcine skin membranes were dosed within 30 min of humane euthanasia of the porcine skin donor, so transepidermal water loss and transepidermal electrical resistance skin integrity tests were not conducted [43]. However, leak tests and visual assessments of damage that may have occurred during the mounting procedure were performed prior to the start of the experiment.

The dermal side of the skin disks was perfused with a bovine serum albumin collection medium and maintained at a pH between 7.3 and 7.6. The temperature of the diffusion cells was maintained at 37 °C ± 1 °C using a heating block. The flow rate was maintained at 4 mL/h using a peristaltic pump. This flow rate was chosen to maintain sink conditions, ensuring that the concentration of the test substances in the receptor fluid remained low, thereby driving continuous diffusion through the skin. The flow rate of 4 mL/h is consistent with standard protocols and regulatory guidelines. The room temperature and relative humidity were recorded throughout the experiment for record keeping. Perfusate samples were collected in glass scintillation vials at 0, 15, 30, 45, 60, 75, 90, 120, 180, 240, 360, and 480 min. After the flow-through diffusion cell systems were set up, the chemical doses were added to each cell. The time for the experiment and sample collection started immediately after the last cell was dosed.

### 2.4. Dosing Procedure

The test chemical ^14^C-phenanthrene was taken from the stock solution and made into a dose mixture. The dosing solution was prepared by adding phenanthrene (15% *v*/*v*) to the artificial sweat (84% *v*/*v*) and then adding acetone (1% of total volume). The dose mixture was vortexed to ensure the test compound was thoroughly mixed. Skin disks were dosed with 100 µL of the phenanthrene mixture to the top of the skin membrane, administering 9.03 µg/cm^2^ (~1.8 µCi). Dose was determined by taking the average of 3 pre- and post-dose aliquots of 10 µL from the dosing solution. After dosing, diffusion cells were covered with Parafilm^®^ pieces (Pechiney Plastic Packaging, Chicago, IL, USA) to minimize the loss of the test compounds. Fifteen minutes after chemical dosing, 100 µL of decontamination wipe solution was applied to the skin surface. Diffusion cells were covered with new pieces of Parafilm. All pieces of Parafilm were collected and saved for ^14^C analysis.

### 2.5. Sample Analysis

Perfusate samples were taken at time points 0, 15, 30, 45, 60, 75, 90, 120, 180, 240, 360, and 480 min after dosing. After the experiment, aliquots of the perfusate were transferred to new scintillation vials along with 15 mL of BioScint (National Diagnostics, Atlanta, GA, USA) and analyzed using a liquid scintillation counter for ^14^C determination. At the end of the experiment, the remaining dose was removed from the surface of the skin membrane with a cotton swab. The skin membranes were transferred to wax paper, where the surface of each skin disk was then tape-stripped (Scotch Tape; 3M, Hutchinson, MN, USA) six times, placing three strips into a single scintillation vial, and adding 10 mL of ethyl acetate. After tape-stripping, the center of the skin disks was punched with an 8 mm biopsy tool, and the center and peripheral skin were separated and placed into individual scintillation vials along with 2 mL of BioSol. The skin samples were incubated at 50 °C for 8–12 h and analyzed using a liquid scintillation counter for ^14^C determination. The fingertips of the gloves used during swabbing and tape stripping were extracted with ethanol.

### 2.6. Absorption Calculations

Absorption was defined as the total percentage of dose detected in the perfusate. Cumulative absorption (µg/cm^2^) was calculated by summing the total dose that was detected in the perfusate at each sampling time. Flux (µg/cm^2^/h) was obtained from the steady-state slope of the cumulative absorption versus time curves; an example is provided in Figure 1. The permeability coefficient (K_p_) (cm/h) was calculated from the ratio of the flux (µg/cm^2^/h) to the concentration (C_s_) (µg/cm^3^) of the dose. The dose concentration was obtained from 10 µL pre- and post-dose checks. The lag time (τ) was obtained by extrapolating the steady-state portion of the curve back to the time- or x-axis. This lag time was related back to diffusivity (D) and membrane thickness (L) by the following equation: D = L^2^/6(τ). Student’s *t*-test was performed to determine significant differences at *p* < 0.05.

## 3. Results

### 3.1. Dermal Absorption, Flux, Diffusivity, Permeability, and Lag Time

The cumulative absorption (µg/cm^2^) of phenanthrene when the wipe solution was used was generally similar, on average ranging within 0.09–0.16 ug/cm^2^ for all wipe solutions tested. Looking at the absorption profiles in Figure 2, wipe solutions 1 and 3 behaved similarly and wipe solutions 2 and 4 also behaved similarly. Although the absorption was similar across all wipe solutions, significant differences were observed between wipe solutions 1 vs. 2 and 2 vs. 3 (Student’s *t*-test *p* < 0.05). The absorption efficiency (% dose absorbed) of phenanthrene was 0.96 ± 0.31, 1.76 ± 0.65, 0.98 ± 0.64, and 1.57 ± 1.05 for wipe solutions 1, 2, 3, and 4, respectively. The cumulative absorption and absorption efficiency were significantly lower (70–86%) when wipe solution was added compared to no wipe solution, illustrated in Figure 2. Flux (µg/cm^2^/h) and permeability (cm/h) decreased when wipe solution was applied to the skin surface vs. no wipe solution. The lag time (minutes) of phenanthrene was reduced from 183.0 ± 20.5 when no wipe solution was applied down to 90.0 ± 17.9, 157.9 ± 32.6, 131.2 ± 48.1, and 164.3 ± 27.4 for wipe solutions 1, 2, 3, and 4, respectively.

Wipe solution pairing 1 and 3 had no significant differences for cumulative absorption, absorption efficiency, lag time, flux, diffusivity, or permeability (Student’s *t*-test *p* < 0.05). No significant differences for the absorption characteristics were also observed between wipe solutions 2 and 4 (Student’s *t*-test *p* < 0.05). Although the absorption efficiency of phenanthrene was slightly different between the two pairings (<1% dose), it does suggest that differences in the wipe solution ingredients may impact dermal absorption. This is supported by the significant differences in absorption efficiency, flux, and diffusivity between wipes 1 and 2 as well as wipes 1 and 4. A summary of the absorption characteristics can be found in Table 2. Overall, the absorption was lower when wipe solution was applied to the surface of the skin compared to when no wipe solution was applied. Data used to generate Figure 2 can be found in Appendix A.

### 3.2. Skin Dispossition and Mass Balance

Percent dose values of phenanthrene detected in the stratum corneum, skin, and skin surface are provided in Table 3. Data used to generate Table 2 and Table 3 can be found in Appendix A. More of the dose was collected from the remaining dose on the skin surface when wipe solutions were applied to the skin than not, i.e., 75–87% vs. 56%, respectively. However, significantly more dose was found present in the stratum corneum for all wipe solutions: 4.0–18% dose vs. no wipe solution, which had a 2.3% dose in the SC. Although significantly higher amounts of phenanthrene were found present in the SC, there was significantly less remaining in the skin (1.0–1.8% vs. 6.8% dose). Lastly, absorption was drastically reduced for all wipe solutions. Overall recovery of phenanthrene was positive, with recoveries greater than 94%.

Although greater amounts of phenanthrene were found in the SC, the wipe solutions reduced the penetration of phenanthrene into the lower layers of the skin. A likely explanation for the greater amount of phenanthrene remaining in the SC could be the wipe solutions hydrating the skin, swelling the SC cells and intracellular keratin, thus allowing for greater skin penetration [20].

## 4. Discussion

Skin decontamination wipes are intended to remove chemical and particulate contamination from firefighter gear and skin. However, firefighters have expressed some concerns over skin decontamination wipes increasing dermal absorption through the “wash-in” phenomenon or the ingredients added to the wipes. These concerns are valid as studies have shown continued absorption of PAHs and chemical warfare agents after a soapy water wash was used to clean the skin [44,45,46]. Furthermore, various decontamination wipes include ingredients such as alcohol derivatives, skin moisturizers, and plant/oil extracts, which have been shown to be potential or known penetration enhancers [47,48,49]. Chemical penetration enhancers increase penetration across the skin by different mechanisms of action: (1) disruption of the intercellular lipid structure between corneocytes in the SC, (2) interactions with intercellular domain of protein, (3) increasing the partitioning of a drug through solvent SC interactions, and (4) enhancers acting on desmosomal connections between corneocytes or altering metabolic activity within the skin [47].

Upon inspection of popular decontamination wipe ingredient lists, there appear to be common ingredients in multiple products. These ingredients include glycerin, aloe vera extract and other plant extracts, caprylyl, tea tree oil, sodium hydroxide and sodium benzoate, polysorbate 20, and citric acid. Out of this list there are four ingredients that have been previously studied for their ability to enhance dermal absorption of drugs or chemicals: glycerin, aloe vera extract, tea tree oil, and polysorbate. Nakashima and coworkers (1996) found that glycerin significantly enhanced the absorption rate of cyclosporin three times at concentrations of 6% (*v*/*v*) dose solution [35]. Mohammadi-Samani and coworkers showed that polysorbate 80 and polysorbate 20 at different concentrations and at various mixture ratios had no detectable penetration-enhancing effects on lidocaine in the guinea pig model [37]. Aloe vera gel at a 0.75% dose solution (*w*/*v*) increased skin penetration of ketoprofen by 2.5 times [50]. Tea tree oil decreased skin integrity in a dose-dependent manner, increasing the absorption of water by 20% in human skin at 5% of pre-treatment volume [51]. Eucalyptus was shown to increase the permeability coefficient of 5-fluorouracil 30–60-fold, but only 1.5-fold for carvedilol [48,52,53]. Furthermore, many creams, gels, and other solvents are frequently used as penetration enhancers; however, depending on their interaction with the penetrating chemical they may decrease absorption. This penetration reduction effect was illustrated with peppermint reducing the absorption of benzoic acid in a dose-dependent manner [51]. As for the other popular skin decontamination wipe ingredients, there is little to no previous literature on their penetration enhancement effects. The exact ratio of skin decontamination wipe solution ingredients is not provided by wipe manufacturers due to proprietary formulation reasons. Regardless it is recommended that wipe manufacturers do not increase the amount of potential dermal absorption-enhancing ingredients until an investigation into the minimum concentration required to illicit an enhancement effect on dermal penetration is conducted.

In general, dermal absorption is a simple diffusion process that depends on concentration gradients but can be complex due to dilution effects based on the penetrating molecule, solvent, and skin conditions [54]. The results in this study showed that all skin decontamination wipe solutions decreased the amount of phenanthrene that was absorbed and remained in the skin. Conversely, higher amounts of phenanthrene were found in the stratum corneum compared to when no wipe was used in Probert et al. (2024) (manuscript submitted for publication) [39]. The reduced amounts of phenanthrene found in the skin and collection media, and higher amounts in the SC, show that the wipe solution reduced the penetration of phenanthrene into the lower layers of the skin.

The reduced skin penetration is likely due to a dilution effect when the skin decontamination wipe solution was applied. Water and soapy water skin decontamination methods have been shown to partially or completely remove chemical warfare agents, PAHs, pesticides, and other contaminants [55,56,57,58,59]. However, in vitro “wash in” effects were observed by Zhu et al., (2016) and Thors et al., (2020) [45,46], whereas Loke et al., (1999) and Forsberg et al., (2020) demonstrated enhanced penetration rates momentarily, but no overall increase in the cumulative amounts penetrated [58,59]. The primary ingredient in each skin decontamination wipe solution is water. Schenk et al., (2018) demonstrated that diluted mixtures generally have a lower flux (mg/cm^2^/h), often one-tenth the flux of neat chemicals [60]. In this study, the addition of skin decontamination wipe solution decreased the flux of phenanthrene by 72–87%, which is similar to the data on organic solvents reported by Schenk et al., (2018) [60].

However, one study conducted by Keir et al., (2023) showed that although water and water–soap decontamination strategies are capable of removing significant amounts of PAHs from the skin, they were unable to significantly reduce PAH internal dose [57]. The findings by Keir et al., (2023) suggest that dermal absorption that may occur during fire suppression activities and cannot be reduced through post exposure skin decontamination. Additional chemical protection in firefighter turnout gear may be necessary to effectively mitigate dermal absorption during fire suppression activities. The findings by Keir et al., (2023) should be reproduced to confirm the potential effectiveness of skin decontamination strategies at reducing firefighter internal doses. Regardless, this does not reduce the importance of post-exposure skin decontamination. The purpose of post-exposure skin decontamination is to minimize the prolonged absorption into the skin after the initial exposure. Furthermore, four studies evaluated time until skin decontamination and each study found that a shorter time until decontamination significantly improved decontamination efficacy [58,59,61,62]. Again, this reiterates the importance of a first-pass skin decontamination after fire suppression activities have concluded.

Among the four decontamination wipes tested, two groups formed between wipes 1 and 3 and wipes 2 and 4. Although the differences in cumulative absorption (µg cm^−2^) between the groups were slight, there were significant differences. Even though there is a slight difference in absorption between the wipes it does suggest an ingredient effect. Further examination into the minimum effective concentration for decontamination wipe ingredients may be warranted. However, even if there are some ingredients that may increase dermal absorption, they are not used at high enough concentrations to illicit a drastic effect, as seen in the comparison to the no-wipe scenario. Overall, the more toxic PAHs have higher molecular weights and are more lipophilic. The results from this study would suggest that the concentration of the more lipophilic PAHs would be diluted when a decontamination wipe is used and decrease firefighters’ dermal absorption of these compounds.

The goal of this study was to determine if the solution in skin decontamination wipes is capable of enhancing the skin penetration of phenanthrene. Experimental conditions were intended to mimic the sweaty conditions of the skin after fire response; this was accomplished by using an artificial sweat dosing vehicle. The skin decontamination wipe solution applied to the skin post-chemical exposure was intended to mimic a firefighter using a wipe to clean their skin during on-scene decontamination. However, there are limitations within this experiment that must be highlighted. The primary limitation of this study is the lack of a water-only control to compare to the effects of the wipe solutions on dermal absorption. Having a water-only control would help determine if the decrease in dermal absorption was due to a dilution effect or the ingredients in the wipe solutions. Furthermore, in vitro studies have repeatedly reported lower absorption compared to in vivo studies across multiple species including human and pig [25,26,44]. Although no animal skin model can exactly replicate the absorption in human skin, pig or porcine skin is the most relevant animal model and has been reviewed and validated over many years [20,63]. The method of using liquid PAHs rather than vapor, particulate, or soil to simulate chemical exposure does not directly represent firefighter exposures. Dosing with liquid phenanthrene was chosen due to greater accessibility and the lack of radiolabeled vapor, particulate, or soil materials. Additionally, the duration for which the skin decontamination wipe solution remained in contact with the skin, i.e., eight hours, is unlikely to occur in the field. Firefighters would be expected to clean the skin immediately after fire exposure, where skin decontamination would be completed within minutes, and further decontaminate the skin by showering upon returning to the fire station. Lastly, the design of this study is unable to include the mechanical forces used in wiping to remove chemicals. Repeating this experiment with the entire skin decontamination wipe would help understand the effectiveness of wipes at removing fireground contaminants from skin.

## 5. Conclusions

Firefighters are routinely exposed to PAHs and other fireground contaminants, which have been found on the gear and skin post-fire exposure. The aim of this study was to investigate if the ingredients in the skin decontamination wipe solution had any penetration enhancement effects on the dermal absorption of phenanthrene, a fireground contaminant. The results of this study show that the skin decontamination wipe solution had no penetration enhancement effects. Rather, the opposite was observed, i.e., the absorption of phenanthrene was decreased with the addition of skin decontamination wipe solution. For firefighters, these results show that the ingredients in the skin decontamination wipes are safe. Skin decontamination wipes should serve as a minimum level of on-scene decontamination for smaller or rural fire departments who may be limited by resources for more sophisticated forms of on-scene decontamination.

The scope of this study may be further expanded in future research to include naphthalene and benzo[a]pyrene. Naphthalene, a lower-molecular-weight PAH, is commonly found in air samples and has been shown to readily penetrate the skin. Benzo[a]pyrene, a high-molecular-weight PAH, is a known carcinogen and has been found on the gear and skin of firefighters. Understanding the effectiveness of skin decontamination wipes at removing these chemicals would better characterize firefighters’ dermal exposures to PAHs and predict the removal of the other priority PAHs. Additionally, rerunning this experiment with a water-only solution would give insight into reducing dermal absorption of fireground contaminants through dilution.

Small differences in absorption were observed between wipe solutions 1 and 3 vs. 2 and 4, indicating that one or more ingredients may impact absorption of phenanthrene. Specific conclusions cannot be drawn from these results, as the ingredients in the wipe solutions were tested as a mixture. To further investigate this finding, the most likely absorption enhancement wipe solution ingredients should be tested individually to determine what concentrations are needed to illicit increased dermal absorption to a degree that would be detrimental to firefighters, thus providing wipe manufacturers with a guide for developing more effective skin decontamination wipe solutions.

Wipe manufacturers should continue to develop, work with, and refine their additive solutions in their skin decontamination wipe products. As shown in this study, the solution in the wipes has the potential to reduce the absorption of chemicals through the skin. The primary mechanism that reduced phenanthrene absorption is believed to be due to diluting the chemical on the surface of the skin. Wipe manufacturers should continue using water as the primary ingredient in their skin decontamination wipe solutions. Continued testing and evaluation of new skin decontamination wipe solutions formulations will be necessary to ensure the continued safety of firefighters.

## Figures and Tables

**Figure 1 toxics-12-00716-f001:**
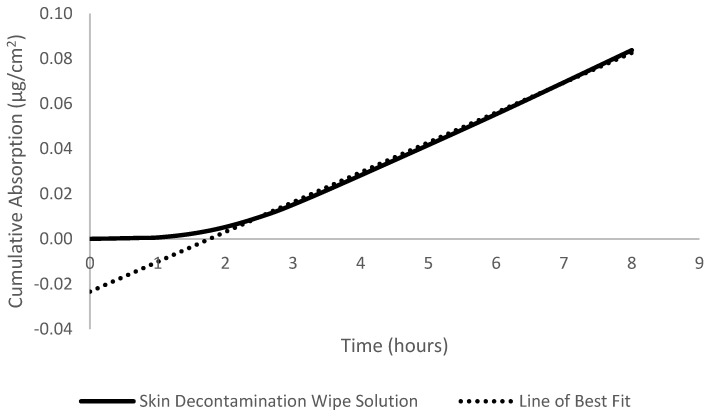
Cumulative absorption (µg/cm^2^) versus time (hours) plot for phenanthrene + wipe 2 solution in an artificial sweat vehicle in porcine skin in vitro flow-through diffusion cells. The best-fit line was used to calculate the flux.

**Figure 2 toxics-12-00716-f002:**
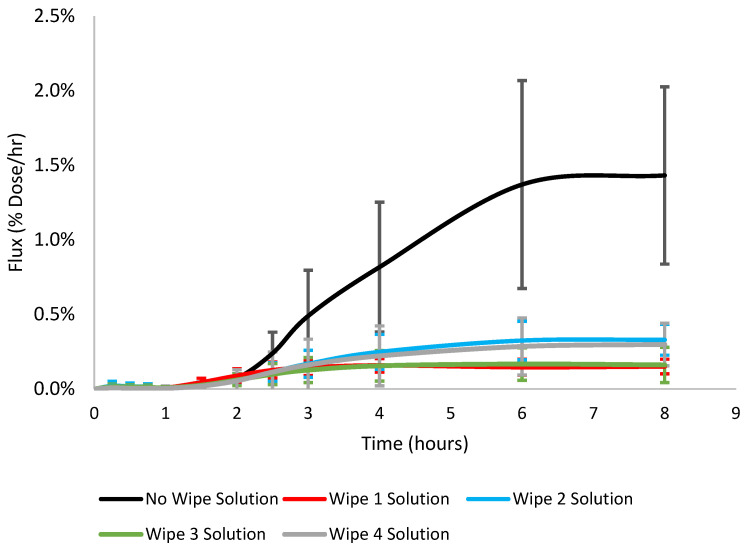
Flux (% Dose/h) profiles of phenanthrene with different skin decontamination wipe solution ingredients (Wipe 1—Red, Wipe 2—Blue, Wipe 3—Green, Wipe 4—Grey) in an artificial sweat dosing vehicle in porcine skin in vitro. The No Wipe absorption profile was generated from data from Probert et al., (2024) [39].

**Table 1 toxics-12-00716-t001:** Ingredient lists of different decontamination wipe products.

Wipe 1	Wipe 2	Wipe 3	Wipe 4
Water (Aqua)	Water	Deionized Water	Water
Hexylene Glycol	Propanediol	Gluconolactone	Phenoxyethanol
Glycerin	Aloe Barbadensis Leaf Extract	Decyl Glucoside	Decyl glucoside
Sodium Hydroxymethylglycinate	Chamomila Recutia (Matricaria) Flower Extract	Sodium Benzoate	Tetrasodium Glutamate Diacetate
Citric Acid	Cucumis Sativis (Cucumber) Fruit Extract	Dehydroacetic Acid	Sodium Benzoate
Disodium Cocoamphodiacetate	Althaea Officinalis Root Extract	Calcium Gluconate	Sodium Citrate
Fragrance	Avena Sativa (Oat) Kernel Extract	Caprylic/Capric Triglyceride	Citric Acid
Sodium Benzoate	Decyl Glucoside	Chamomilla Recrutia Extract	Sodium Bicarbonate
Potassium Sorbate	Polyglyceryl-10 Caprylate/Caprate	Aloe Barbadensis Extract	Glycerin
Disodium Cocoyl Glutamate	Coco Glucoside	Tocopheryl Acetate (Vitamin E)	Tocopheryl Acetate (Vitamin E)
Sodium Cocoyl Glutamate	Glyceryl Oleate		Cucumis Sativus (Cucumber) Fruit Extract
	Polysorbate 20		
	Tetrasodium Glutamate Diacetate		
	Trisodium Phosphate		
	Citric Acid		
	Caprylyl glycol		
	Benzalkonium Chloride		
	Sodium Benzoate		
	Potassium Sorbate		
	Phenoxyethanol		

**Table 2 toxics-12-00716-t002:** Absorption characteristics of phenanthrene in porcine skin in vitro dosed in an artificial sweat dosing vehicle with skin decontamination wipe solution applied post-chemical dose (mean ± standard deviation).

Decontamination Wipe	Cumulative Absorption (µg/cm^2^)	Absorption Efficiency (% Dose)	Flux (µg/cm^2^/h)	Lag Time (minutes)	Diffusivity (cm^2^/h)	Permeability (cm/h) (×10^−4^)
Wipe Solution 1	0.09 ± 0.03	0.96 ± 0.31	0.013 ± 0.005	90.0 ± 17.9	0.63 ± 0.15	2.3 ± 0.8
Wipe Solution 2	0.16 ± 0.06	1.76 ± 0.65	0.029 ± 0.010	157.9 ± 32.6	0.36 ± 0.06	5.1 ± 1.8
Wipe Solution 3	0.09 ± 0.06	0.98 ± 0.64	0.015 ± 0.009	131.2 ± 48.1	0.46 ± 0.12	2.6 ± 1.6
Wipe Solution 4	0.14 ± 0.09	1.57 ± 1.05	0.026 ± 0.016	164.3 ± 27.4	0.35 ± 0.06	4.5 ± 2.7
No Wipe Solution *	0.53 ± 0.25	6.80 ± 3.20	0.104 ± 0.045	183.0 ± 20.5	0.20 ± 0.02	21.0 ± 9.1

* Data from Probert et al., (2024) [39].

**Table 3 toxics-12-00716-t003:** Phenanthrene recovered from absorbed, skin, stratum corneum, and skin surface as a measure of percent dose (mean ± standard deviation).

Wipe	Dose (µg/cm^2^)	Remaining Dose (% Dose)	Stratum Corneum (% Dose)	Skin (% Dose)	Absorption (% Dose)	Total Recovery (% Dose)
Wipe Solution 1	9.03	86.8 ± 5.2	4.0 ± 4.7	2.9 ± 1.5	1.0 ± 0.3	94.6 ± 2.3
Wipe Solution 2	9.03	81.7 ± 5.6	8.3 ± 9.5	4.0 ± 4.0	1.8 ± 0.6	95.8 ± 2.7
Wipe Solution 3	9.03	75.9 ± 11.1	18.1 ± 14.0	4.0 ± 1.0	1.0 ± 0.6	99.0 ± 4.2
Wipe Solution 4	9.03	82.2 ± 7.8	8.7 ± 10.2	2.8 ± 2.5	1.6 ± 1.1	95.3 ± 3.0
No Wipe Solution *	7.76	56.5 ± 3.5	2.3 ± 0.6	32.4 ± 5.2	6.8 ± 3.2	98.1 ± 3.3

* Dose and mass balance data are from Probert et al., (2024) [39].

## Data Availability

Data is contained within the article or Appendix A.

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
