# Peer review of "Impact of Skin Decontamination Wipe Solutions on the Percutaneous Absorption of Polycyclic Aromatic Hydrocarbons"

_toxics, 2024, doi:10.3390/toxics12100716_

Round 1

Reviewer 1 Report

Comments and Suggestions for Authors

The study addresses an important topic but appears to have a significant methodological flaw and is presented in such a fashion that it could be misinterpreted by the reader.  Furthermore there is no limitations section.

Title: The title should be changed to "Impact of Decontamination Wipe Solutions on the Percutaneous..." as the wipes themselves were not directly tested. 

Abstract: Line 14, please change to "the effect of decontamination wipe solution..."as again the aim of your study was to test solution, not the wipes themselves. Please look for similar issues throughout the manuscript.

Introduction: In the third paragraph on page 2 you mention decontamination of gear and skin decontamination.  These should be presented separately or mention of gear contamination removed.  

Methods: It appears that the control treatment did not involve addition of  100ul of solution, and given the minimal differences between the various solutions, the main reduction in absorption could have been entirely due to dilution (as suggested in the discussion) and not to any of the chemicals in the wipe solution.  If 100ul of distilled water or other similar chemical free solution was added to the no-wipe control, then please clearly state it.  Otherwise the study should be rerun to address this limitation.  

Results: For Figure 1 and Tables 2 and 3 the legend or column title should state "Decontamination Wipe Solution", not "Decontamination Wipe". On line 194 "significant" is used but no statistical test is mentioned.

Discussion: On line 256 "all wipes" should be changed to "all wipe solutions". A limitations paragraph should be added as it is not currently present. The final sentence on likely differences by lipophilicity are not supported by the study or otherwise referenced.  I suggest that the lack of skin wipe efficacy in preventing systemic absorption, as shown by Keir et al., be prominently mentioned here.

Conclusions: The first sentence is not supported by the study and should be deleted. The section should start with "The ingredients in the decontamination wipe solution..."

Reviewer 2 Report

Comments and Suggestions for Authors

The aim of the manuscript entitled "Impact of Decontamination Wipes on the Percutaneous Absorption of Polycyclic Aromatic Hydrocarbons" is to investigate the impact of different decontamination wipes on the percutaneous absorption of phenanthrene, a polycyclic aromatic hydrocarbon commonly found in firefighting environments. Specifically, the study seeks to determine whether these wipes enhance or reduce dermal absorption of phenanthrene by evaluating various absorption characteristics, such as cumulative absorption, absorption efficiency, lag time, flux, diffusivity, and permeability, using an in vitro flow-through diffusion cell system.

The relevance of the topic today is high, as occupational safety and health for firefighters remain a critical issue. With increasing awareness of the long-term health impacts of carcinogen exposure, there is a growing emphasis on developing and implementing effective decontamination strategies. Additionally, advancements in decontamination technology and products make it essential to assess and validate their efficacy continually. The manuscript is within the journal's scope. It is very well prepared in general. Still, some minor issues need to be addressed. My specific comments are given below.

The abstract could be strengthened by a more straightforward conclusion that briefly states the implications of the findings, such as which types of ingredients in wipes might be more effective or safer for use by firefighters.

The introduction section should provide more comprehensive background information on the influence of PAHs on human health.

The authors should claim the novelty of the manuscript at the end of the Introduction.

To improve the conclusion, the authors should emphasize the practical implications of immediate decontamination for firefighter safety and clarify the significance of the findings regarding the lack of penetration enhancement by wipes. They should also expand on future research directions by specifying which PAH compounds or contaminants should be tested next and stress the need for detailed analysis of wipe ingredients to determine their effects and optimal concentrations. Additionally, the authors should refine recommendations for wipe manufacturers to focus on using the study's findings to develop safer and more effective decontamination products.

Comments on the Quality of English Language

Minor changes are required. 

Round 2

Reviewer 1 Report

Comments and Suggestions for Authors

The authors have made edits addressing most of the commments.  For the study abstract, the sentence "This study was aimed to understand the effect of decontamination wipes..." should be changed to "This study was aimed to understand the effect of decontamination wipe solution..."

Author Response

The manuscript has been revised to better communicate the testing of wipe solutions and wipe solution ingredients in the sections detailing the aims and goals of the manuscript. 

Reviewer 2 Report

Comments and Suggestions for Authors

The authors addressed all my comments. 

Comments on the Quality of English Language

Minor changes are required. 

Author Response

Reviwer 2 Comment: The authors addressed all my comments. Minor editing of English language required. 

Author Response: Thank you. We have gone through the manuscript to improve the language of the manuscript to better read for English readers. 

We have also included response for Reviewer 1 comments as there were comments remaining to be addressed but no separate link to address them. 

Reviewer 1 Comments: “It appears that the control treatment did not involve addition of 100 µL of solution, and given the minimal differences between the various solutions, the main reduction in absorption could have been entirely due to dilution (as suggested in the discussion) and not to any of the chemicals in the wipe solution. If 100 µL of distilled water or other similar chemical-free solution was added to the no-wipe control, then please clearly state it. Otherwise, the study should be rerun to address this limitation."

Author Comments: Can be found in the attachment. 
